# The Adverse Environmental Impact Factors Analysis on Fly-In-Fly-Out Personnel at Industrial Enterprises

**DOI:** 10.3390/ijerph19020997

**Published:** 2022-01-17

**Authors:** Yana Korneeva

**Affiliations:** Department of Psychology, Northern (Arctic) Federal University, 163002 Arkhangelsk, Russia; amazonkca@mail.ru; Tel.: +7-960-004-5657

**Keywords:** differential analysis, adverse environmental impact factors, fly-in-fly-out work, industrial enterprises, group isolation, north, south, industrial psychology

## Abstract

(1) Background: the research purpose is to identify and describe the general and different factors of adverse environmental impact on FIFO personnel at industrial enterprises at different levels of differential analysis of professional activity. (2) Methods: The research involved 359 employees of industrial enterprises with FIFO work organization. The study was carried out using a questionnaire, including a subjective assessment of the discomfort of three groups of negative environment impact factors to the FIFO personnel: climatic-geographical, industrial and social. (3) Results: The relationship between the increase in the degree of discomfort of production factors due to the in-fluence of climatic, geographical and social conditions has been established. With a various location of objects, the greatest discomfort is felt from the action of climatic and production factors; with varying degrees of group isolation and the shift period duration—all three groups, with the greatest influence of domestic and social; in various industries and enterprises—all three groups. (4) Conclusions: The differential analysis of the professional activities of FIFO personnel of industrial enterprises should be carried out at the following levels: the location of an industrial facility, the degree of group isolation, the duration of the shift period, the industry, the type of enterprise and the professional group.

## 1. Introduction

The fly-in-fly-out (FIFO) employment continues to expand at the present time. The FIFO method of organizing work is denoted by the terms “Fly-in-Fly-out”, “long distance labor commuting” or “shift work” and is understood as “any job in which work is so isolated from the places of permanent residence of workers that they food and housing are provided at the workplace, and lists are created according to which workers spend a fixed number of days at the site (shift camp), followed by a fixed number of days of rest at home” [1] (p. 2). This method is used to service all remote production areas, including both offshore oil and gas fields and onshore production projects. It is important to note that the FIFO method is “the only viable option for offshore oil and gas industry, since, in contrast with on-shore mine sites, daily travel to work in these conditions is impossible, and permanent places of residence are not available” [1]. Companies see rotational work as an effective way to meet the growing demand for minerals and an opportunity that also offers several benefits [2].

The use of the FIFO work method is spreading fast to various sectors of economy, including construction, production and transport. Geographically, FIFO personnel have become popular not only in significantly remote areas of the Far North and the Arctic, but also in the central and southern regions. The choice as empirical objects for the study of industrial enterprises are due to a number of reasons. Traditionally, enterprises, organizations and objects at which the FIFO method of organizing work can be applied include companies in the oil and gas industry, forestry, geological exploration and railway transport, as well as construction and installation trusts. A great demand for FIFO workers is observed in the construction industry, in particular, large and new industrial facilities. Manpower resources to ensure full-scale construction are not sufficient in the regions, on the one hand, and the termination of the need for them after the completion of construction, on the other, makes the use of FIFO work forms and the involvement of specialists from other regions of the country urgent.

It has been established in many studies that three groups of factors affect FIFO workers in the Far North and the Arctic: the climatic and geographic factors associated with the harsh Arctic climate; the industrial ones, due to hazardous and harmful nature of production, and the social ones, determined by group isolation conditions. Climate and geographic, production and social factors impose requirements on the organism of a FIFO worker, which often exceeds its reserves, thus excluding the possibility of full adaptation to these conditions and leading to occupational health risks.

Scientific research in the field of studying the impact of factors of the FIFO method on workers was carried out mainly in the Far North and the Arctic. It was revealed that professional activity on a FIFO basis in the Far North contributes to the formation of unfavorable functional states and the development of destructive personal qualities of workers, which leads to a decrease in the mental health level, productivity and efficiency of labor activity.

To date, a large number of studies have been carried out on the development and implementation of biomedical programs and technological and organizational measures, such as a comprehensive system of medical support for FIFO personnel in the Far North and the Arctic [3,4,5,6,7,8] or a system for reducing accidents and injuries of FIFO personnel of oil and gas producing enterprises [9,10,11,12] and the model of medical and psychological support for FIFO work carried out in the Far North [13,14,15,16,17,18,19,20,21]. The wide variability of industries that use the FIFO work method makes it necessary to adapt existing technologies of medical and psychological support for personnel.

The FIFO work method involves working in harsh climatic and geographical conditions and re-quires mobilization of adaptive resources. At the same time, the professional success of specialists is due, first of all, to the success of overcoming environmental limitations (climatic and geographical factors, group isolation with, often, unwanted people, intensive work and rest regimes, harmful and dangerous conditions). Thus, working on a FIFO basis is always stressful, and very few people have sufficient adaptive reserves and effectively spontaneously adapt to such conditions. Some workers cannot cope with the load and quit within 1–2 years, the rest achieve success in their professional activities on a FIFO basis at the cost of excessive consumption of internal resources, which can ultimately lead to a loss of physical and mental health. Consequently, FIFO personnel need comprehensive medical and psychological support. The methodology for such support is still to be developed. We have proposed a meta-adaptive approach using a differential analysis of professional activity [22].

The differential nature of the analysis is due to the fact that it is built as an ascent from the analysis of the most general factors affecting the entire staff of a group of professions, to the study of factors that determine the working conditions of individual specialized groups of professions and only then to the study of factors that determine the working conditions of each individual profession [22]. By de-fining the levels of differential analysis of FIFO personnel’s professional activities, we can, on the one hand, determine general patterns in the dynamics of workers’ states, their health status, psychological well-being, efficiency and safety of the production process, and, on the other hand, identify important differences between groups, and thereby predict professional and psychological risks of personnel at various production facilities.

The aim of our research is to identify and describe the general and different factors of adverse environmental impact on FIFO personnel at industrial enterprises at different levels of differential analysis of professional activity: at the level of climatic, industrial and social factors.

The research hypothesis: based on the substantiation of the levels of differential analysis of the professional activity of FIFO personnel of industrial enterprises (p. 3), we assumed that differences in the subjective assessments of the discomfort of environmental factors among the FIFO personnel at industrial enterprises will be observed at all differential analysis levels. At the same time, large differences will be observed in the subjective assessments of climatic and geographical factors at the levels of the region where the industrial facility is located, the fly-in period duration and the group isolation degree; in assessments of social and household factors at the levels of the fly-in period duration and the group isolation degree, and production factors at the level of industries.

To determine the levels of differential analysis of the FIFO work, it is necessary to analyze the key climatic, geographic, production and social factors affecting workers of various industrial facilities.

## 2. The Substantiation of the List of Unfavorable Environmental Factors for FIFO Personnel at Industrial Enterprises

### 2.1. Climate and Geographic Factors of Adverse Environmental Impact on FIFO Workers at Industrial Enterprises

As stated earlier, most FIFO work facilities are located in the extreme climatic and geographical conditions of the high latitudes of the Far North and the Arctic. These territories are characterized by the following climatic features: intense wind regime (due to high cyclonic activity) with high humidity in combination with low temperature, sharp changes in barometric pressure, significant seasonal photoperiodicity, including polar day and polar night phenomena [23,24,25,26,27,28]. This leads to a significant tension in the regulatory systems of the human body, causing the development of various diseases: climatic and geographical factors are primary importance for the cardiovascular and musculoskeletal systems diseases, ENT organs, social factors play a significant role in the occurrence of the gastrointestinal tract diseases and infectious diseases [29,30,31,32,33,34,35,36,37,38,39,40,41,42].

The above climatic and geophysical conditions cause the development of northern stress or “the polar tension syndrome” in the population, which is characterized by the following components: oxidative stress, insufficiency of detoxification processes and barrier organs, disorders of the northern type of metabolism, northern tissue hypoxia, immune deficiency, blood hypercoagulation, polyendocrine disorders, functional dissymmetry of interhemispheric relationships, desynchronosis, psychoemotional stress, meteopathy, etc. [4,23].

Changes in meteorological conditions in the Far North, including a significant change in air temperature and significant drops in atmospheric pressure, are reflected, first of all, in such manifestations of stress as an increase in psycho-emotional stress and an increase in the level of meteosensitivity [41]. Climatic and weather factors have an adverse effect on human activities: they lead to interruptions in telecommunication and navigation systems, which poses danger to astronauts, aircraft crews and passengers; they disrupt the operation of oil and gas equipment, railway transport, etc. [42]. The health and well-being of workers is also affected.

Other researchers came to similar conclusions regarding the following unfavorable climatic factors [23,24,25,26,27,28]: low air temperature, high humidity, disproportionate ratio of dark and light time of the day (polar night and polar day) and increased wind regime.

One of the most significant factors is ambient air temperature. As we move to the North, low air temperatures increase the primary disease incidence of the population, especially the diseases of the upper respiratory tract. Pronounced drops in pressure and density of atmospheric oxygen in combination with low air temperatures and strong winds cause excessive stress on the bronchopulmonary system of the body [39].

According to Veremchuk and Kiku, high humidity and strong winds, being the most unfavorable factors in the development of respiratory diseases, increase the cooling capacity of the air [40].

A number of researchers note a direct dependence of the incidence of cardiovascular diseases on unfavorable meteorological factors [39,41,42].

Increased irritability, depression, insomnia, affective reactions and aggressiveness in the inhabitants of the north are caused by a constant unusual light regime, fog, precipitation, low temperatures and cold winds even in summer.

During the polar night, constant darkness, low temperatures, high humidity, strong winds, sharp weather changes in short periods of time and magnetic storms provoke a decrease in the functional state of people, expressed in an alarming alertness to weather, exaggeration of difficulties, anxious and suspicious attitude to health, decreased efficiency, drowsiness and fatigue [43].

Tucha described the relationship of the negative impact of natural and climatic conditions of human life on labor intensity in the mining regions of the North [44].

Thus, it became possible to identify the key climatic and geographic factors affecting FIFO workers in the Far North and the Arctic:-low temperatures,-high humidity,-polar night,-polar day,-change of time zones,-geomagnetic disturbances,-wind,-insufficient ultraviolet radiation,-the radiation situation,-barometric pressure drops,-change in oxygen content in atmospheric air.

Due to the fact that in this study some of the industrial facilities are located in the south of the Russian Federation, after analyzing additional sources, we supplemented the list of climatic and geographical factors with the following ones [45,46]:-high temperatures,-drought,-rains/showers,-dust storms.

In the present study, in all samples, the discomfort of the presented factors was assessed by employees in order to identify the differences and determine the degree of their adverse impact and the employees’ adaptation level.

### 2.2. Industrial Factors of Adverse Environmental Impact on FIFO Workers at Industrial Enterprises

With regard to production factors, the impact of different modes of work and rest on FIFO personnel has been studied widely, but this issue remains controversial. One and the same work and rest regime in different territories (in different latitudes) and at different enterprises can vary significantly in terms of the characteristics of the influence [47].

Features of work and rest regime affect health and psychological condition of FIFO personnel, which is reflected in the studies on the physiological parameters of the state of the body [48] and the quality of sleep, its effectiveness and duration [49]. It was found that the greatest harm to the body is caused by a daily shift schedule of 12 h of work/12 h of rest [48]. Similar data on the negative impact of work and rest regime 12/12 were obtained in the studies of FIFO workers of an offshore oil platform in the North Sea [50].

The negative impact of night shifts on employees should be taken into account: recovery from night shifts is the worst. However, after a week of restorative rest in the inter-shift (fly-out) period (after a 14-day fly-in period), sleep is fully restored.

In addition, the number of fly-in days is of particular importance. The organization of work during the day also matters. For example, a 12-h shift has an extremely heterogeneous dynamics of working capacity and functional state, therefore, the organization of work must take into account this dynamic in order to avoid health problems and injuries. Traditionally accepted fly-in-fly-out modes in Western Europe are 2 weeks of work and 2 weeks of rest, the mode of 3 weeks of work and 3 of rest, or 2 weeks of work and 3 weeks of rest is used much less often. A combination of alternating periods of work—rest 2/2 and 3/3 is also being tested. Studies show that the most optimal mode in this case is 2 weeks of work and 3 weeks of rest, but this schedule reduces FIFO work economic feasibility.

Research by Parkes on 17 offshore platforms is aimed at studying various factors: working conditions, composition, personal attitude to work, general health and satisfaction at different enterprises. The author found statistically significant differences between drilling rigs and offshore platforms (*p* < 0.001). The conditions at offshore platforms were assessed as more favorable, and the working regime was more comfortable. With regard to employment at these two sites, the following features were identified: approximately 75% of the rig personnel worked in a drilling company, 30% of the platform production personnel were employees of contractors [51].

Safety culture as a risk reduction factor. There is a need to strengthen work in certain areas, such as the development of safety culture standards; harmonization of safety standards obtained in practice with the organization of various processes in the industry, increasing the competence of employees and expanding their participation in safety activities and decision-making [52].

According to Alekseenko, production factors have a statistically significant influence on the health of shift workers. He also pointed out that an increase in the degree of harmfulness of vibration, noise, microclimate and the severity of the labor process causes an increase in the incidence of sickness among engineering and technical personnel for servicing drilling rigs [53].

Of particular importance is the cumulative effect of production factors and conditions of a job post that are contrasting in the degree of action and nature. Labor tasks of workers of various industries differ in physical and physiological stress, as well as in the degree of action of harmful production factors [54].

Compared to onshore FIFO workers, offshore workers experience increased levels of anxiety, more sleep problems and more stress. Some potential stressors for offshore personnel include limited living and working conditions, lack of privacy, constant noise and activity, complex shift mechanisms with workloads ranging from periods of boredom to periods of focused activity [55,56].

Berezin and Vyzhigin studied combined harmful effects on workers in the oil fields of the Far North caused by physical and chemical factors of the working environment and factors of the labor process. With the interaction of climatic and production factors, working conditions at 75% of workplaces are recognized as harmful or dangerous. The most common factors are noise, lighting, vibration and microclimate [57].

Rustamov et al. mentioned the following unfavorable labor factors affecting personnel in oil production: dynamic and static loads, forced working posture, high neuro-emotional tension, rotational work organization, impractical work mode, rest and nutrition [58].

Safonova found that in the diamond mining industry, working conditions, along with the existing harmful factors (dust, gas, vibration and noise), are complicated by exposure to low temperatures and the presence of chemicals. Working conditions, the nature of professional activity, track record in the North and living conditions affect workers’ disease incidence [59].

Based on the foregoing and on the results of numerous studies of harmful and hazardous factors of extractive industries and construction [4,21,31,48,49,50,57,59,60,61] we have identified a list of key production factors affecting FIFO personnel:-noise,-vibration,-chemical factors,-illumination,-physical exercise,-monotony and static physical activity,-increased surface temperature of equipment,-high and low temperature of the working area,-air humidity in the working area,-static electricity level,-prohibition to leave the premises during working hours,-neuropsychic overload,-enhanced control over compliance with corporate regulations,-enhanced monitoring of compliance with safety regulations,-no choice of short breaks,-no professional support from colleagues,-work at height,-high intensity of activity.

The present study also assessed the degree of discomfort of these factors as seen by FIFO workers at industrial facilities.

### 2.3. Social and Household Factors of Adverse Environmental Impact on FIFO Workers at Industrial Enterprises

The “FIFO lifestyle” significantly changes the social situation of human development. The frequency of work activities performed at a high pace and in stress, alternating with long rest and physical relaxation, changes a person’s life, his attitude to life circumstances and the circle of contacts. The peculiar biological conditions of this activity change one’s endurance to physical and mental stress. The author proposes to use anxiety as an indicator of emotional stress and as an indicator of the level of mental adaptation to various types of activity [62].

Davydova identified the main features of social interaction within a team of FIFO workers, which makes it possible to define this team as a small group [63]. She also developed a system of indicators and methods for measuring parameters of the activity of a rotational team, highlighted specific, empirically verified parameters that affect the social and economic efficiency of activities (presented in Figure 1).

The following peculiarities of the life of FIFO workers with a negative impact include:-group isolation and forced circle of contacts;-narrowing down of personal space, which makes it impossible to withdraw oneself from the others;-differences in life values, differences in views on the life of workers of different ages;-decrease in the information background of communication, information depletion;-lack of development of cultural infrastructure;-family responsibilities are performed by additional personnel (cleaning the premises, cooking, etc.) [64,65].

The living conditions at the rig (everyday discomfort) do not presuppose a separate room for each employee. The dwelling accommodates at least 2 people. The living space is very small. While it is possible to have solitary walks in the summer months during the inter-shift (fly-out) period, this becomes impossible during the polar night with its severe weather; therefore, the 12-h rest also takes place in a confined space in the company of other people. When a person knows that he is being watched or there is a chance that it might happen, then he constantly tries to perform a certain role function, which causes emotional tension. A person is forced to suppress his true feelings and desires [66].

An important reason for the growth of emotional tension is information depletion, restriction of the inflow of personally significant information, “information hunger”. It arises due to the monotony of the social circle over a long period (14 or more days) during the shift period. The FIFO method specificity involves the group isolation of collectives, limitation of the telephone communication and the Internet, associated with climatic and geographic factors (the location of objects in the Arctic far from settlements).

Working on a rotational basis makes various and sometimes unusual requirements to employees, which may be associated with the following features [62]:-a change in the usual daily sleep patterns—wakefulness, uniqueness of nutrition, sanitary and hygienic conditions, unusual conditions of rest, sensory deprivation;-specific social and psychological conditions (communication with a fairly narrow circle of people; isolation from the usual family life; unusual situation, lack of the possibility of complete solitude);-work in hazardous conditions.

One of the most unpleasant consequences of the impact of negative factors of the FIFO work organization is the subjective feeling of social isolation. Recent advances in communication (e.g., video calling, social media) can help reduce, but not completely mitigate, some of the geographic distance issues for shift workers and their partners [67].

Another negative factor of working on a rotational basis, due to group isolation, are “situations when close relatives have problems at home, and you cannot help them”. The partners noted that one of the most negative aspects of working on a FIFO basis is the concern that employees will not be able to return home in case of personal need. A previous study found that 80% of a sample of Australian FIFO workers reported that their employer was flexible about taking time off in the event of a family crisis [68].

Group isolation can be defined as “a forced long stay of a group of people in a limited space, scarcity of sensory stimuli and constant communication with the same people” [62,63,66,69]. Isolation of a person or a group of people from usual living conditions and communication with other people during a space flight, wintering, remote expeditions, etc.

The psychogenic factors of group isolation, which determine its specificity, include constant “publicity” and informational monotony in communication. The need to keep oneself in a certain role position for a long time, the desire to hide one’s thoughts and feelings from others, and the resulting emotional tension make the need for solitude more urgent [69].

Social desynchronosis (term Garanina’s term) also belongs to social limitations A person who works in a shift schedule mode has two temporal concepts (“systems of generalized temporal representations” of the personality): general social and specific individual, due to the peculiarities of the work schedule (“internal” desynchronosis). There is also a mismatch between the lifetime of the specialist himself and the lifetime of other people (significant others) working by the usual schedule (“external” desynchronosis)” [70]. The phenomenon of “social desynchronosis” arises as a result of the need to constantly switch from one image to another [70].

The level of social dissatisfaction of FIFO workers can be reduced with the help of a number of management decisions. The relatively high level of wages and the social package of enterprises are a significant factor in minimizing the level of turnover in northern collectives. However, negative aspects of FIFO life accumulate and often manifest themselves in a form of interpersonal conflicts between employees. Maslakov’s polls made it possible to identify the level of conflict in FIFO teams and its causes, identified by the participants in the conflicts themselves. The following most important reasons were noted: dissatisfaction with wages, accrual of bonuses; dissatisfaction with the conditions of industrial life; dissatisfaction with the organization of production and labor; the shift rotation mode; dissatisfaction with working conditions [71].

The change in the composition of workers (during shifts), including direct managers, is accompanied by a negative impact on the information field of the FIFO team. Interaction requires communication and exchange of information, which must be started from the beginning if there is a change in communication participants.

The research demonstrates the importance of factors in the design of the object of lifework for productivity growth and reduction of the number of employee errors [72,73]. The quality of the accommodation of FIFO workers, the arrangement of rooms for sleeping, eating and resting, affects work productivity and the overall sense of comfort and well-being. As the cumulative effect of the impact of environmental factors on shift personnel is large, it is important to correctly design the premises for work and housing for employees, taking into account the impact of the environment. Similar requirements are necessary for the crews of ships that spend a long time at sea, and their level of performance directly depends on the conditions. In order to improve the comfort level, the American Bureau of Shipping ABS Plaza has created a special guide to regulate the arrangement of these conditions [73].

Based on the foregoing and on research on social and household factors of FIFO work, caused by group isolation [62,63,64,66,67,68,69,70,71,72], we have identified a list of key social and domestic factors affecting FIFO personnel, which were assessed by employees according to the degree of their discomfort:-social isolation of the team,-unfavorable accommodation conditions,-lack of food,-difficulty with transport and communication,-psychophysiological and psycho-emotional discomfort,-frequency of work,-forced circle of contacts,-lack of developed infrastructure,-narrowing the personal space zone,-information exhaustion,-lack of a developed cultural infrastructure,-ban on smoking and alcohol,-lack of psychological support from colleagues.

## 3. Substantiation of the Determination of the Levels of Differential Analysis of the Professional Activity for FIFO Personnel at Industrial Enterprises

When developing the level of differential analysis, we relied on the specifics of FIFO work organization. According to our analysis of previous research [74], three groups of environmental factors influence workers in rotational forms: climate and geographic, industrial and social. The variability of the action of each group of the presented factors varies depending on the specific enterprise, at the same time, in each of these groups, its own criteria can be distinguished, through which the degree of environmental impact can be differentiated. Figure 2 shows the levels of differential analysis of professional activity with the FIFO method of labor organization, according to which we generalize the data of our research.

At the level of climatic and geographical factors, this is the region where the industrial facility is located (north or south). Previous studies have established the negative impact of extreme climatic and geographical conditions of the Far North and the Arctic. At the same time, due to the prevalence of the rotational method in southern regions, it is necessary to take into account the specifics of climatic and geographical factors in these territories (for example, extremely high air temperatures, showers, dust storms, etc.).

At the level of social conditions, due to group isolation, it is necessary to take into account the degree of group isolation and the length of the FIFO period.

Simonova [66] studied the effect of the group isolation degree on the psychological characteristics of FIFO personnel in the exploration and oil and gas production in the Arctic and found out that the relevant groups were differentiated by the group isolation degree as follows: (1) workers who work and live in the shift camp (about 200 people)—a weak isolation degree; (2) oil production area (30–40 people)—medium degree of isolation; (3) workers employed on remote drilling rigs (about 15 people)—a strong isolation degree.

It should be noted that when studying the group isolation degree at mining and construction sites, it is necessary to differentiate the weak degree using the other factors. In these conditions, there are separate cases (such as an oil and gas production operators), when professional activities are carried out individually, and after the shift employees return to the rotational camp. Therefore, we have presented the division of objects according to the group isolation degree, using two criteria: the number of people simultaneously living in the shift camp and the remoteness of the object from the settlement. As shown earlier, group isolation is characterized by information depletion and the variety of information received. The quality of communication depends both on the number of employees living in the rotational camp and on the variety of means of communication with the environment (which is directly related to the remoteness of the object and the specifics of its activities).

Table 1 shows the characteristic of study samples by differential analysis levels. As noted, a low group isolation degree is noted during the construction of the Crimean Bridge, since the rotational camp is located within the boundaries of the settlement. After the shift employees can leave the camp and visit nearby settlements, while the number of employees simultaneously living in the settlement exceeds 1000.

A high group isolation degree is noted among representatives of the offshore oil production platform, which is associated, on the one hand, with a small number of people simultaneously present, limited space (offshore facility), as well as limited means of communication at the facility (severe limitation of mobile communications and open access to the Internet), due to the requirements of secrecy and security, on the other hand.

The construction and oil production facilities located in the Arctic have the above average degree of group isolation. This is due to a relatively small number of employees living simultaneously in the shift camp (about 300 people), as well as recurrent problems with communication and transport, caused by the extreme climatic and geographical conditions of the land part of the Arctic territories.

The average degree of group isolation is noted at a diamond mining facility, where, provided that employees are simultaneously close to facilities with an isolation degree above average, good communication with the outer world, both through mobile communication and the Internet. These opportunities appeared due to the location of the industrial facility in the more favorable climate conditions of the Far North.

The importance of assessing the impact of the fly-in period duration on the professional efficiency, safety, health and psychological well-being of FIFO personnel is shown in the works of different researchers [47,48,49,50]. In our study, all employees are divided into groups with a fly-in periods of 14 days, 28 days, and over 28 days.

## 4. Materials and Methods

Research type is empirical, analytical, transverse. All research methods were considered at the ethics committee of the Higher School of Psychology, Pedagogy and Physical Culture of the Northern (Arctic) Federal University and recommended for use (protocol No. 3, 2017).

### 4.1. Sample

To achieve this goal, a study was carried out during five scientific expeditions to the following industrial facilities with a FIFO work organization:(1)construction of main gas pipelines, Republic of Komi (Far North); 82 employees took part (average age 34.91 ± 0.926, average FIFO work experience 4.57 ± 0.343, fly-in period—52 days);(2)oil production, Nenets Autonomous Okrug (Far North); 67 employees took part (average age 38.46 ± 1.410, average FIFO work experience 9.85 ± 1.072, fly-in period—28 days);(3)diamond mining production, Arkhangelsk region (Far North); 77 employees took part (average age 38.56 ± 1.151, average FIFO work experience 8.16 ± 0.701, fly-in period—14 days);(4)construction of the “Crimean Bridge” facility, Krasnodar Territory (south); 83 employees took part (average age 41.31 ± 1.242, average FIFO work experience 8.02 ± 0.721, fly-in period—28 days);(5)offshore ice-resistant oil production platform, Caspian Sea (south); 50 employees took part (average age 36.17 ± 1.064, average FIFO work experience 7.97 ± 0.839, fly-in period—14 days).

The total sample size is 359 employees of industrial enterprises with FIFO work organization. The participation in the study was voluntary (all the participants signed a written voluntary informed consent). In terms of the education level, all surveyed employees were distributed as follows: 14.2%—general secondary education; 52.8%—secondary vocational education; 7.9%—incomplete higher education; 25.1%—higher education.

From the point of view of production factors, all industrial facilities are divided by industry (oil and gas production, diamond mining and construction), by specific enterprises and enlarged professional groups.

In Figure 3 shows the distribution of the research samples by the levels of differential analysis of FIFO work. These data confirm the diversity presented and the consideration of the main factors in the formation of research samples, and also allows us to determine the sufficiency of empirical data for comparative studies at each level of differential analysis.

The research was carried out through scientific expeditions, in which there was a stay of the research psychologist during the entire shift (fly-in) period of the personnel. Employees were asked to answer a questionnaire during their stay at an industrial facility and to assess the degree of unfavorable environmental factors discomfort.

### 4.2. Procedure

The research was carried out using a questionnaire, developed by us, which included the following sections: general information about education and work experience; marital status; subjective assessment of the discomfort of climatic-geographical, industrial and social factors that affect workers during the shift arrival; features of the organization of free time during the fly-in period; subjective assessment of professional efficiency; subjective assessment of the danger of various situations that may arise during the fly-in period; subjective assessment of the hazard in the workplace and the factors contributing to its emergence. All assessments of the studied parameters were made on point and rank scales.

The subjective assessment of the discomfort of climatic, geographical, industrial and social and household factors that affect workers during the fly-in period was carried out on a five-point scale, where 1—the factor is not experienced, the person has adapted to its action, and 5—the factor is experienced as most uncomfortable. Individual question weights were considered equal and no alpha coefficients were calculated. This is also due to the fact that the sample is a priori inhomogeneous in terms of the weight contributions to the factors taken into account.

The list of factors that were subject to assessment is presented in Figure 4 and in Section 2.1, Section 2.2 and Section 2.3.

For the most part, we are talking about the adverse impact on workers, which is expressed both in a direct negative impact and in what creates stress for workers. Stronger control implies high penalties for non-compliance and slightest assumptions on employees, which creates additional tension and stress. Since this model of factors was tested for several years, these factors were identified in addition to those proposed by employees of enterprises, as those that cause additional discomfort. Therefore, they were also included in the final model.

The study was complex in nature, including observation of employees, conversations, psychophysiological and psychological testing. This article presents some of the results of the work obtained on the basis of a survey of employees.

### 4.3. Data Analysis

Statistical analysis was carried out using descriptive statistics, multivariate analysis of variance (MANOVA) and correlation analysis using Pearson χ^2^. The MANOVA method was used because the study of the relationship of many factors was carried out.

For all parameters presented in this research, the criterion of equality of the variances of Levene’s errors is greater than 0.05. Before the analysis, the data were checked for normality of distribution for the possibility of using MANOVA. The statistical normality test of Kolmogorov-Smirnov is used.

Processing was carried out using the SPSS 23.00 software package (license agreement No. Z125-3301-14, IBM, Moscow, Russian Federation).

Research stages:(1)comparative analysis of the factors of unfavorable impact of environment on FIFO personnel at different geographic locations of industrial facilities;(2)comparative analysis of the factors of unfavorable impact of environment on FIFO personnel at different degrees of group isolation of teams;(3)comparative analysis of the factors of unfavorable impact of environment on FIFO personnel at different duration of the shift period;(4)comparative analysis of the factors of unfavorable impact of environment on FIFO personnel working in various industries;(5)interrelation of the factors of unfavorable impact of environment on FIFO personnel with different levels of differential analysis of professional activity.

To achieve the goal of this study, firstly, we analyzed the differences in the assessments of three groups of factors (climate and geographic, production and social factors) separately at each level of differential analysis, then, for the visual study of the obtained relationships at all levels, a correlation analysis was carried out, followed by the construction of the correlation constellation.

## 5. Results

### 5.1. Comparative Analysis of the Factors of Unfavorable Impact of Environment on FIFO Personnel at Different Geographic Locations of Industrial Facilities

To identify the differences in the subjective assessment of the discomfort of unfavorable environmental factors felt by FIFO workers at different geographical locations of industrial facilities, three multivariate analyses of variance (MANOVA) were successively applied, where the fixed factor was the attribution of an industrial facility in the north or south of the Russian Federation. The dependent variables were as follows: in the first multivariate analysis of variance—parameters of climatic and geographical factors; in the second analysis—the parameters of production factors; in the third—the parameters of social factors.

Bock’s M is 53.8 at *p* = 0.105, hence the variance-covariance matrices are not statistically significantly different and the main assumption for multivariate tests is fulfilled. According to the data of multivariate criteria, there are statistically significant differences between representatives of the southern and northern watches in the estimates of climatic and geographical (Pillai Trail 0.174, F = 8.304 at *p* ˂ 0.001) and production (Pillai Trail 0.178, F = 3.062 at *p* ˂ 0.001) factors, relatively social—household (Pillai Trace 0.049, F = 1.211 at *p* = 0.270) factors, statistically significant differences are not observed. Before the analysis, the data were checked for normality of distribution for the possibility of using MANOVA.

The validity of assessing the differences between groups using univariate criteria for assessing the discomfort of social conditions can be assessed according to the correction for the multiplicity of comparisons. Let us set the border of the error of the first kind—0.05/13 = 0.004. The next social and living factor, according to the univariate criteria for comparing the groups of the southern and northern shifts, crossed the threshold of significance of 0.004: lack of food (descriptive statistics for groups and the significance of differences are presented in Table 2).

As shown in Table 2, low or high air temperatures and wind are the most unfavorable factors for employees in the north and south of the Russian Federation. Moreover, higher estimates of temperature discomfort are noted among FIFO workers in the south. This is due to the fact that at extremely high air temperatures in the south, both work premises and equipment are heated to the maximum, which complicates the performance of work and negatively affects the condition of workers. This is confirmed by higher estimates of employee discomfort in the south of such production factors as increased surface temperature of equipment and higher and lower temperatures of the working area.

The presence of wind increases the uncomfortable effect of low temperatures. As a result, the personnel at the northern industrial facilities give higher estimates of the significance of this factor. As shown earlier, a characteristic feature of the climate in the North is a sharp drop in barometric pressure and a change in the oxygen content in the atmospheric air, due to which we observe higher estimates of the discomfort of these factors among FIFO personnel at facilities located in the Far North and in the Arctic.

With regard to production factors, it should be noted that the estimates of physical activity and noise by employees of the southern regions are higher, which is due to the specifics of the activity.

Thus, our analysis showed that at the facilities located both in the north and in the south of the Russian Federation, there are special climatic and geographic factors that have a negative impact on personnel. These factors enhance the effect of production environmental factors and add the new ones, since a number of works are carried out in the open air or in the cabin of universal equipment.

### 5.2. Comparative Analysis of the Factors of Unfavorable Impact of Environment on FIFO Personnel at Different Degrees of Group Isolation of Teams

As noted earlier, according to the group isolation degree, all employees are divided into four groups. A low group isolation degree presupposes the location of an industrial facility within settlements, the number of people in the shift camp is more than 1000 people, and the availability of all means of communication with the external environment. The average group isolation degree assumes the remoteness of the object by 130 km from settlements, the presence of no more than 350 people in the shift camp at the same time, the availability of telephone and Internet communications during the rest period. Above average, the group isolation degree is characterized by a distance from settlements of 150 km or more, no more than 370 people living in a shift camp of, limited means of communication due to the extreme climatic and geographical conditions of the Far North. A high group isolation degree implies a distance of about 180 km from settlements, no more than 105 people staying at the facility and in a residential block at the same time and restrictions on telephone and Internet communication with the outer world due to security requirements and corporate standards.

To identify differences in the subjective assessment of the discomfort of unfavorable environmental factors on FIFO workers with a different degree of group isolation of teams, three multivariate analyses of variance were sequentially applied, where the fixed factor was assigning a low, medium, above average or a high degree of group isolation to an industrial facility. Dependent variables were as follows: in the first multivariate analysis of variance—the parameters of climatic and geographical factors; in the second analysis—the parameters of production factors; in the third—the parameters of social factors. M Boxa 46.2 at *p* = 0.342, therefore the variance-covariance matrices do not differ statistically significantly and the main assumption for multivariate tests is fulfilled. According to multivariate criteria, there are statistically significant differences between groups with varying degrees of group isolation in assessments of climatic and geographic (Pillai Trail 0.424, F = 6.498 at *p* ˂ 0.001), industrial (Pillai Trail 0.468, F = 2.811 at *p* ˂ 0.001) and social household (Pillai Trace 0.359, F = 3.166 at *p* ˂ 0.001) factors. According to Scheffe test, significant differences are observed in the assessment of social factors between groups with low and above average group isolation.

As can be seen from the data in Table 3, differences in assessments between employees working in different group isolation conditions are observed for all social factors. At the same time, the highest estimates of discomfort are noted among employees with a degree of group isolation above average, i.e., at facilities located in the Arctic. These features are influenced by the geographical distance and the severity of climatic conditions, which necessitates additional measures to ensure the comfort of the personnel during the fly-in period of the personnel.

It should be noted that the next group giving the maximum assessments of the discomfort of social and living conditions are employees with a high degree of group isolation. It should be noted that they carry out their activities in the south of the Russian Federation, nevertheless, they note unfavorable living conditions, information depletion, and social isolation. This confirms our assumption that the isolation degree of teams is a significant factor influencing the performance and psychological well-being of employees. It can be complicated by an additional increase in the duration of the fly-in period and extreme climatic conditions.

At the same time, influencing the development of tension, psychological discomfort, it can increase the sensitivity of personnel to the current climatic and industrial factors. As can be seen from these estimates of production factors, physical activity and air humidity in the working area are assessed as more uncomfortable by employees with a high degree of group isolation. High ambient temperatures, and high humidity (among climatic factors) also have a more pronounced significance for them compared to other groups.

Employees with a degree of group isolation above the average assess more negatively (in comparison with others) the effect of geomagnetic disturbances, wind, time zone changes, barometric pressure drops and changes in the oxygen content in the atmospheric air, and among industrial ones—the effect of chemical factors and illumination. This also indicates an increased sensitivity to their action in conditions of a higher isolation degree and a longer fly-in period. For a group with an average group isolation degree and the location of the facility in the Far North, the effect of these climatic factors is also relevant, but employees assess them as comfortable.

### 5.3. Comparative Analysis of the Factors of Unfavorable Impact of Environment on FIFO Personnel at Different Duration of the Shift Period

To identify differences in the subjective assessment of the discomfort of unfavorable environmental factors on FIFO workers with different fly-in period duration, three multivariate analyses of variance were sequentially applied, where the fixed factor was attribution to an industrial facility with a different fly-in period duration (2 weeks, a month, or more than 1 month). Dependent variables were as follows: in the first multivariate analysis of variance—parameters of climatic and geographical factors; in the second analysis—the parameters of production factors; in the third one—the parameters of social factors. Box’s M 64.2 at *p* = 0.411, therefore the variance-covariance matrices do not differ much statistically and the main assumption is fulfilled for multivariate tests. According to multivariate criteria, there are statistically significant differences between groups with varying degrees of group isolation in assessments of climatic and geographic (Pillai Trail 0.232, F = 5.178 at *p* ˂ 0.001), industrial (Pillai Trail 0.308, F = 4.991 at *p* ˂ 0.001) and social household (Pillai Trace 0.397, F = 5.771 at *p* ˂ 0.001) factors. According to Scheffe test, the most excellent group in assessing social and living factors is a group with a shift period of more than 1 month.

Table 4 presents statistically significant differences according to univariate tests in the subjective assessment of the discomfort of climate and geographic, production and social factors in FIFO personnel with a different fly-in period duration.

As in the previous block, employees with a different fly-in period duration do not have uniform assessments of the discomfort of all socio-psychological factors. At the same time, as can be seen from the data in Table 4, employees who work in the mode for more than 1 month give higher marks in all parameters. This confirms the assumption of the cumulative effect of the negative impact of the environment at a long duration of the rotational drive.

As with the group isolation degree, the fly-in period length can increase the effect of climatic, geographical and production factors on employees. In confirmation of this, it should be noted that there are higher estimates of the discomfort of high humidity, change of time zones, geomagnetic disturbances, wind, changes in barometric pressure and changes in oxygen content in the atmospheric air among shift workers with a shift duration exceeding one month.

Greater discomfort of production factors is noted among employees with a rotational arrival duration of 28 days, only with respect to vibration, the ratings are higher for a group with a 14-day shift (which we associate with the specifics of the activity).

### 5.4. Comparative Analysis of the Factors of Unfavorable Impact of Environment on FIFO Personnel Working in Various Industries

To identify the differences in the subjective assessment of the discomfort of unfavorable environmental factors on FIFO workers in various industries, three multivariate analyses of variance were successively applied, where a fixed factor was attribution to an industrial facility in various industries (diamond mining, oil and gas production, or construction). Dependent variables were as follows: in the first multivariate analysis of variance—the parameters of climatic and geographical factors; in the second analysis—the parameters of production factors; in the third one—the parameters of social factors. Box’s M is 51.3 at *p* = 0.155, hence the variance-covariance matrices are not much different statistically and the main assumption is fulfilled for multivariate tests. According to the data of multivariate criteria, there are statistically significant differences between groups with different degrees of group isolation in the assessments of climatic and geographic (Pillai Trail 0.171, F = 3.696 at *p* ˂ 0.001), industrial (Pillai Trail 0.230, F = 3.559 at *p* ˂ 0.001) and social household (Pillai’s trace 0.323, F = 4.489 at *p* ˂ 0.001) factors. According to Scheffe test, the most excellent group for assessing social and living factors are representatives of construction.

Among representatives of various industries, we expected to receive the maximum differences in the estimates of production factors (Table 5). For diamond workers, vibration is more uncomfortable, while for oil and gas workers, these include chemical factors, light and physical activity. Builders appreciate the effect of physical activity, elevated surface temperature of equipment, air humidity and lack of professional support from colleagues as uncomfortable as possible.

Among climatic factors, employees of the oil and gas industry give the maximum assessment to low or high temperatures, wind and barometric pressure drops, while builders—high humidity, geomagnetic disturbances, radiation conditions and changes in oxygen content in the air. Employees of the diamond mining enterprise assess most of the climatic factors as comfortable ones, attaching importance only to low temperatures and wind.

From the point of view of social arrangement of life during the shift period, the most unfavorable conditions from the point of view of employees are the conditions at construction sites. Most often, during construction, employees live in accommodation cabins, because there is not always a need for infrastructure development on projects under construction (for example, during the construction of gas pipelines). This can be the reason for the problems with becoming established at a new place and low grades for comfort given by employees.

At oil, gas, diamond and other production facilities, employees carry out work at one facility for a long time, as a result of which comfortable living conditions are created for personnel.

### 5.5. Interrelation of the Factors of Unfavorable Impact of Environment on FIFO Personnel with Different Levels of Differential Analysis of Professional Activity

To determine the relationship of the combined influence of factors of adverse environmental impact with different levels of differential analysis of the professional activity of FIFO personnel, a correlation analysis was used, after which a correlation pleiad was compiled (Figure 5). In the Pleiad, the correlations from 0.15 were considered at *p* = 0.047, due to the fact that for a sample of 359 or more, the Pearson coefficient was 0.15 [75,76]. A table with correlation values and significance levels is presented in Appendix A.

On the correlation pleiad (Figure 5), all previously analyzed connections are clearly presented. It was found that at each level of differential analysis the connections with the corresponding group of factors were established. Thus, the attribution to a certain region of the location of an industrial facility has reliable connections with estimates of the discomfort of climatic and geographical factors, and the maximum connections are observed with low/high air temperatures, changes in oxygen content in the air, as well as with a number of production factors. This confirms the idea that extreme climatic conditions of professional activity enhance the effect of production factors.

The group isolation degree and the fly-in period duration have statistically reliable connections (at the level of ≤0.001) with all social conditions, which confirms the significance of these integral characteristics for the peculiarities of the implementation of professional activities on a FIFO basis. The intensity of their impact is also determined by climatic and geographical conditions, which is confirmed by the presence of a relationship with estimates of the discomfort of the polar day, wind, changes in oxygen content in the air, etc.

Most of the relationships with production factors were obtained at the level of industries, specific enterprises and professional groups. At the same time, these characteristics are influenced by climatic and social factors. In connection with the fact that differential analysis goes from general to specific, at its lower levels we see a relationship with all the listed groups of factors. During the analysis at the first levels (the region where the industrial facility is located, the degree of group isolation, the duration of the shift period), we can differentiate more clearly the actions of a specific group of factors.

## 6. Discussion

Thus, the degree of discomfort of climatic, geographic, industrial and social factors makes it possible to assess the level of adverse environmental impact on personnel at each level of the differential analysis of professional activity. The relationship between the increase in the degree of discomfort of production factors due to the influence of climatic, geographical and social conditions has been established. With a different geographical location of objects, the greatest discomfort is felt from the action of climatic and production factors; with varying degrees of group isolation and the duration of the shift period—all three groups, with the greatest influence of domestic and social; in various industries and enterprises—all three groups, with the greatest industrial influence; with various professional groups—industrial and social.

Our study has confirmed the data that unfavorable climatic and geographic factors of both the Far North and the South enhance the effect of production factors [29,39,42,43,46], as well as social conditions [54,56,62,65].

The role of social and living conditions in the shift period on production efficiency and safety is shown [71,72,73].

The duration of the shift period plays an important role and requires ensuring greater comfort of living conditions during the fly-in period, which also confirms the data of other researchers [48,49,50,51]. Our study found that employees who work more than one month give higher scores for all factors of adverse environmental impact. This confirms the assumption of the cumulative effect of the negative impact of the environment at a long duration of the rotational drive.

When working on a rotational basis, an important role is played by ensuring communication with the outer world, especially with relatives and friends, which especially increases with a growth of the fly-in period duration [66,67].

The greatest number of factors with high ratings of discomfort are noted by the bridge builders (noise, chemical factors, illumination, physical activity, elevated equipment surface temperature, air humidity, there is no professional support from colleagues). This may be due to the specifics of the work that is carried out in the open air, and the negative effect of production factors is enhanced by climatic conditions (high humidity, wind, high temperatures), for which employees also give high marks. From the point of view of social and living conditions, the builders of the bridge noted higher ratings of the discomfort of most factors. In this case (the shift camp is located within the boundaries of the settlement), this may be due to the specifics of the corporate culture of the contractors involved.

The second enterprise in which employees give high marks to the discomfort of production factors is an offshore oil and gas production platform. The personnel note the following factors: noise, vibration, physical activity, elevated surface temperature of the equipment and air humidity. This is due to the specifics of this industrial facility. The large number of the production platforms personnel, the location of the facility in the sea at a considerable distance from the coast, changeable weather conditions significantly complicate the process of timely and unhindered evacuation of people in the event of a fire, their protection on the escape routes from the effects of dangerous fire factors and safe escape. platforms in the event of a critical emergency. Another feature of offshore platforms is the maximum degree of utilization of the useful area of the structure, dense placement at all platform levels of a large amount of equipment for various functional purposes. The above factors cause a high likelihood of emergencies.

In addition, these employees give higher ratings to high air temperatures and high humidity. The high isolation of offshore industrial facilities places higher demands on both safety compliance and the identity of the workers themselves. In this case, favorable climatic conditions are not a mitigating factor that provides additional resources, but on the contrary, in situations of high air temperature, they can lead to an increase in the negative effect.

From the point of view of social and living conditions, platform employees note their comfort.

Employees of oil and gas production in the onshore part of the Arctic note chemical factors, illumination, physical activity and noise as the most uncomfortable (compared to other enterprises). Among the climatic factors, geomagnetic disturbances, wind, barometric pressure drops, changes in the oxygen content in the atmospheric air and low temperatures are distinguished, due to the remote location of this facility, the duration of the rotational period and the degree of group isolation, as shown earlier. Among the social factors, the complexity of transport and communication, psychophysiological and psycho-emotional discomfort and the lack of a developed infrastructure are noted.

This study demonstrated the specificity of the negative impact of climatic and geographical environmental factors on FIFO personnel, both in the north and in the south. This requires employers to take all the necessary measures for protecting workers, because the above factors directly affect the production process. Our research has shown the importance of studying not only northern climatic factors for workers, but also southern ones, which have their own specifics.

The obtained results show the highest possible estimates of the discomfort of social and domestic factors among employees with the above average degree of group isolation, i.e., at the facilities located in the Arctic, allow us to draw a conclusion about the influence of geographical remoteness and the severity of climatic conditions on the requirements for increasing personnel’s comfort. This necessitates additional measures for ensuring this during the fly-in period.

A high degree of group isolation, even with a relatively short duration of the fly-in period (14 days) and more comfortable climatic conditions, contributes to a significant increase in the discomfort of living and working conditions. This is mainly due to the limitation of communication with the ex-ternal environment. This emphasizes the importance of communication with family and friends, the creation of the best conditions for this by enterprises. This will significantly reduce the psychological risks of personnel.

In terms of industries, large psychological risks are observed in the oil and gas industry and construction compared to the diamond mining industry.

With regard to the analysis of the results obtained by the levels of differential analysis of the professional activity of the FIFO personnel of industrial enterprises, the following conclusions can be drawn:-at the level of the region where the industrial facility is located—psychological risks exist for the workers both in the northern and southern regions;-at the level of the degree of group isolation—large psychological risks are observed in employees with a high and above average group isolation degree;-regarding the duration of the shift period—with an increase in the duration, there is an increase in the intensity of the impact of environmental factors, especially with a shift period of more than one month;-in relation to industries—large psychological risks for employees of oil and gas production and construction.

When including an enterprise in a group with high psychological risks of personnel at each of the levels of differential analysis, it is necessary to think over and implement additional measures to im-prove the social conditions of employees, as well as reduce the effect of climatic and geographical conditions through the use of PPE and modern technological solutions.

The practical implications. When using the FIFO work method, it is necessary to improve the quality of social and living conditions: the convenience of living, the availability of individual space (by accommodating no more than two people in the room), the presence of good telephone and Internet communication with a family and friends, etc. The higher the degree of group isolation of collectives, these factors play an important role (since their dissatisfaction leads to a greater negative effect of industrial and climatic conditions). There are social factors that allow employees to recover their strength and resources in a timely manner and with high quality, allow them to maintain the required level of working capacity for a long time.

## 7. Conclusions

The differential analysis of the professional activities of FIFO personnel of industrial enterprises should be carried out at the following levels, allowing to maximize the details of adverse impact of climatic, geographical, industrial and social factors of the environment: the location of an industrial facility, the degree of group isolation, the duration of the shift period, the industry sector, the type of enterprise, the professional group, etc.

The levels of differential analysis of the professional activity of FIFO personnel of industrial enterprises are determined taking into account the action of each of the groups of factors of the negative impact of the environment: climatic, geographical, industrial and social, and include the following ones: the region of the industrial facility (north or south), the degree of group isolation (low, medium, above average, high), the duration of the shift period (14 days, 28 days, more than 28 days), industry (oil and gas, diamond mining, construction), enterprise, professional group.

The degree of discomfort of climatic, geographic, industrial and social factors makes it possible to assess the level of adverse environmental impact on personnel at each level of the differential analysis of professional activity. The relationship between the increase in the degree of discomfort of production factors due to the influence of climatic, geographical and social conditions has been established. With a various location of objects, the greatest discomfort is felt from the action of climatic and pro-duction factors; with varying degrees of group isolation and the shift period duration—all three groups, with the greatest influence of domestic and social; in various industries and enterprises—all three groups, with the greatest industrial influence; with various professional groups—industrial and social.

The limitations of this study are associated with a certain list of climatic, geographic, industrial and social factors, which might be detailed and expanded in the future. In addition, the data of this study can be extended only to industrial enterprises, while the FIFO method can be applied in other areas with their own specifics. The expansion of the data obtained by applying the present study design to industrial sites in other countries, taking into account national specificities, may also be expected. The limitation of the research is the geographical location of the objects (north and south), as well as the seasons of the years in which the study was carried out (spring), which should be taken into account when applying the results of this study.

In continuation of this study, we are currently dealing with the features of professional efficiency and personality of employees of all the studied industries at all levels of differential analysis of professional activity. Further, we plan a more detailed development of measures for psychological support of personnel, including their testing at industrial facilities.

## Figures and Tables

**Figure 1 ijerph-19-00997-f001:**
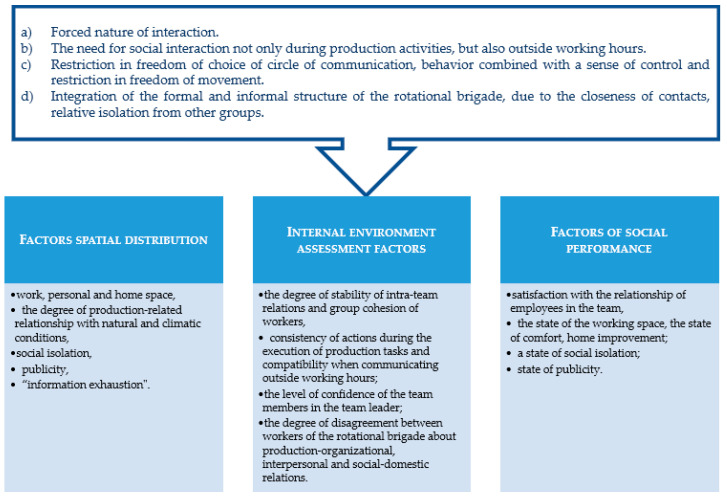
Features of social interaction in the FIFO team and the system of indicators of the parameters of its activities (based on Davydova’s results [63]).

**Figure 2 ijerph-19-00997-f002:**
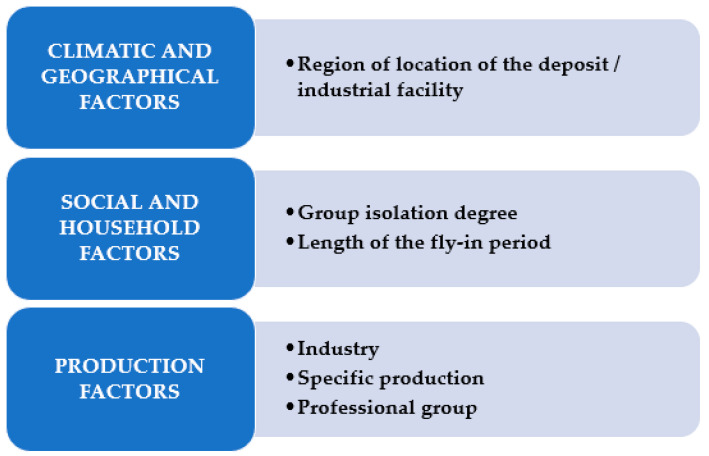
Levels of differential analysis of professional activity for FIFO method of labor organization.

**Figure 3 ijerph-19-00997-f003:**
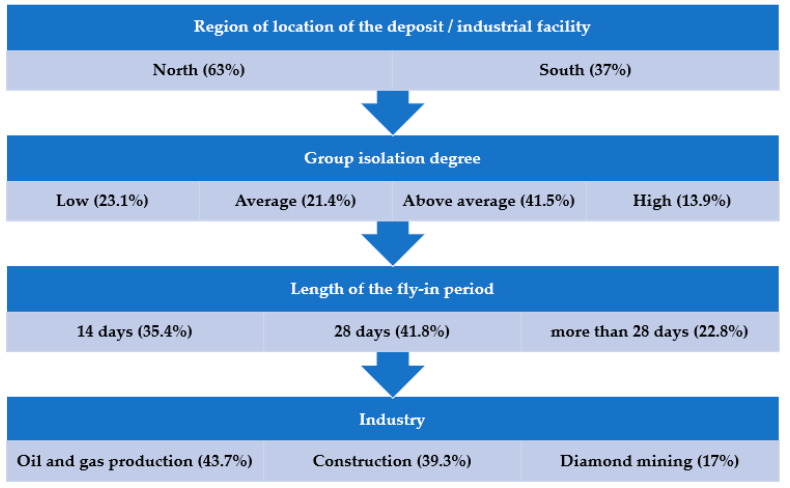
Distribution of research samples by levels of differential analysis of FIFO work. (% of the total research sample). Note: Figure shows how the survey sample is distributed in relation to each of the levels, in parentheses is the% of the sample corresponding to each group.

**Figure 4 ijerph-19-00997-f004:**
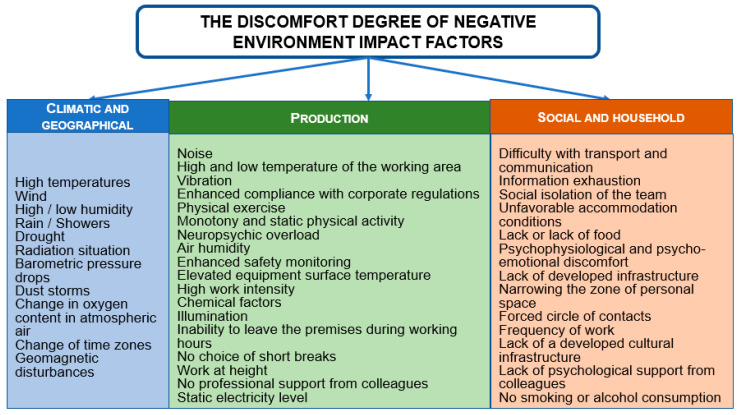
The negative impact of the environment factors lists on the FIFO personnel of industrial enterprises.

**Figure 5 ijerph-19-00997-f005:**
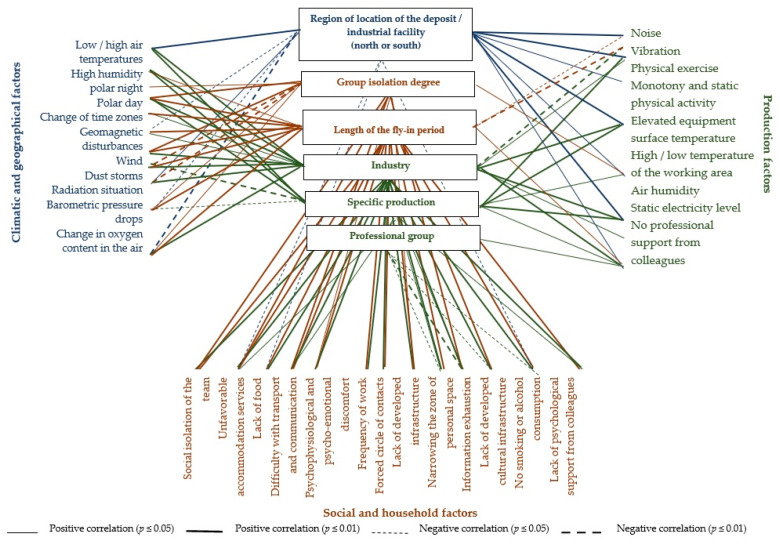
Correlation pleiad of interconnection of the impact of adverse environmental factors with different levels of differential analysis of professional activities of FIFO personnel.

**Table 1 ijerph-19-00997-t001:** Characteristic of samples by differential analysis levels.

Parameter	Construction of Main Pipelines	Oil and Gas Production	Diamond Mining Production	Construction of the Object “Crimean Bridge”	Oil and Gas Production Platform
Geographic location of the object (North or South)	North	North	North	South	South
Number of people in the shift camp	300 people	370 people	305 people	3000 people	105 people
The remoteness of the object from settlements	150 km	311 km	130 km	0 km	180 km
Availability of communication with the outside world and reference groups	low	average	above average	high	low
Group isolation degree	above average	above average	average	low	high
Length of the fly-in period	45 days	28 days	14 days	28 days	14 days

**Table 2 ijerph-19-00997-t002:** Statistically significant differences according to univariate tests in the subjective assessment of the discomfort of climatic, geographical, industrial and social factors among FIFO personnel in the north and south of the Russian Federation (on a five-point scale, where 5 is the most uncomfortable factor).

Factor Name	M ± SD for the North	M ± SD for the South	M ± SD for Entire Sample	Significance Level *
Climate and geographic factors				
Low or high temperatures	2.46 ± 1.349	3.13 ± 1.502	2.73 ± 1.447	0.000
Geomagnetic disturbances	1.56 ± 1.092	1.30 ± 0.889	1.46 ± 1.022	0.028
Wind	2.91 ± 1.428	2.60 ± 1.417	2.79 ± 1.429	0.057
Radiation situation	1.63 ± 1.163	1.95 ± 1.408	1.76 ± 1.274	0.024
Barometric pressure drops	1.79 ± 1.179	1.47 ± 1.024	1.66 ± 1.129	0.012
Change in oxygen content in atmospheric air	1.91 ± 1.284	1.49 ± 1.105	1.74 ± 1.232	0.002
Production factors				
Elevated equipment surface temperature	1.63 ± 1.104	2.18 ± 1.388	1.91 ± 1.285	0.001
Air humidity	1.58 ± 1.070	2.10 ± 1.353	1.84 ± 1.247	0.001
No professional support from colleagues	1.29 ± 0.847	1.69 ± 1.273	1.50 ± 1.102	0.005
Physical exercise	1.82 ± 1.089	2.18 ± 1.247	2.00 ± 1.185	0.015
Noise	2.48 ± 1.395	2.84 ± 1.494	2.66 ± 1.455	0.053
High and low temperature of the working area	2.09 ± 1.255	2.40 ± 1.251	2.25 ± 1.260	0.056
Socially domestic factors				
Lack of food variety	2.34 ± 1.448	2.01 ± 1.290	2.20 ± 1.393	0.040

Note: the table shows the factors for which there are statistically significant differences between the groups. The significance level of *p* is given according to these univariate criteria. *: The significance level is indicated according to tests of between-subjects effects of MANOVA

**Table 3 ijerph-19-00997-t003:** Statistically significant differences according to univariate tests in the subjective assessment of the discomfort of climate and geographic, production and social factors for FIFO personnel with different degrees of group isolation (on a five-point scale, where 5 is the most uncomfortable factor, and 1 is the least uncomfortable).

Factor Name	M ± SD for the Low Isolation Degree	M ± SD for the Average Isolation Degree	M ± SD for a Higher-Than-Average Isolation Degree	M ± SD for a High Isolation Degree	M ± SD for Entire Sample	Significance Level *
Climate and geographic factors						
Low or high temperatures	3.06 ± 1.528	1.91 ± 1.097	2.74 ± 1.382	3.25 ± 1.466	2.73 ± 1.447	˂0.001
High humidity	2.22 ± 1.294	1.63 ± 1.027	2.21 ± 1.297	2.33 ± 1.521	2.11 ± 1.301	0.007
Change of time zones	1.27 ± 0.742	1.15 ± 0.723	1.57 ± 1.145	1.17 ± 0.859	1.35 ± 0.949	0.007
Geomagnetic disturbances	1.43 ± 0.961	1.19 ± 0.609	1.74 ± 1.233	1.08 ± 0.710	1.46 ± 1.022	˂0.001
Wind	2.59 ± 1.412	1.91 ± 1.069	3.43 ± 1.310	2.63 ± 1.438	2.79 ± 1.429	˂0.001
Radiation situation	2.22 ± 1.475	1.40 ± 0.954	1.74 ± 1.245	1.50 ± 1.167	1.76 ± 1.274	˂0.001
Barometric pressure drops	1.44 ± 0.894	1.37 ± 0.832	2.00 ± 1.275	1.50 ± 1.220	1.66 ± 1.129	˂0.001
Change in oxygen content in atmospheric air	1.54 ± 1.107	1.27 ± 0.687	2.25 ± 1.392	1.40 ± 1.106	1.74 ± 1.232	˂0.001
Production factors						
Chemical factors	2.39 ± 1.579	1.61 ± 1.100	2.81 ± 1.415	1.96 ± 1.429	2.18 ± 1.458	˂0.001
Illumination	2.04 ± 1.307	1.49 ± 0.894	2.23 ± 1.323	1.87 ± 1.140	1.90 ± 1.204	0.005
Physical exercise	2.16 ± 1.227	1.54 ± 0.823	2.17 ± 1.279	2.22 ± 1.295	2.00 ± 1.185	0.002
Elevated equipment surface temperature	2.28 ± 1.432	1.46 ± 0.990	1.85 ± 1.211	2.02 ± 1.305	1.91 ± 1.285	0.002
Air humidity	2.05 ± 1.301	1.39 ± 0.953	1.83 ± 1.167	2.18 ± 1.451	1.84 ± 1.247	0.002
No professional support from colleagues	1.78 ± 1.312	1.21 ± 0.686	1.40 ± 1.015	1.53 ± 1.198	1.50 ± 1.102	0.017
Socially domestic factors						
Social isolation of the team	1.79 ± 1.166	1.43 ± 0.908	2.41 ± 1.412	1.96 ± 1.322	1.98 ± 1.296	˂0.001
Unfavorable accommodation conditions	1.74 ± 1.040	1.27 ± 0.750	2.48 ± 1.225	1.86 ± 1.275	1.94 ± 1.193	˂0.001
Lack of food	2.08 ± 1.339	1.85 ± 1.294	2.60 ± 1.463	1.90 ± 1.212	2.20 ± 1.393	0.001
Difficulty with transport and communication	2.06 ± 1.315	1.40 ± 0.818	2.92 ± 1.475	2.47 ± 1.371	2.31 ± 1.421	˂0.001
Psychophysiological and psycho-emotional discomfort	1.81 ± 1.223	1.40 ± 0.780	2.24 ± 1.329	1.88 ± 1.317	1.90 ± 1.239	˂0.001
Frequency of work	1.85 ± 1.080	1.51 ± 0.943	2.12 ± 1.292	1.65 ± 1.071	1.85 ± 1.159	0.003
Forced circle of contacts	1.80 ± 1.216	1.51 ± 0.990	2.08 ± 1.215	1.80 ± 1.274	1.85 ± 1.195	0.015
Lack of developed infrastructure	1.90 ± 1.259	1.51 ± 1.078	2.50 ± 1.330	1.94 ± 1.265	2.05 ± 1.304	˂0.001
Narrowing the zone of personal space	1.91 ± 1.304	1.27 ± 0.770	2.12 ± 1.337	1.86 ± 1.414	1.85 ± 1.277	˂0.001
Information exhaustion	1.79 ± 1.187	1.34 ± 0.770	2.40 ± 1.207	2.14 ± 1.514	1.98 ± 1.243	˂0.001
Lack of a developed cultural infrastructure	2.11 ± 1.484	1.51 ± 0.975	2.63 ± 1.438	1.71 ± 1.307	2.12 ± 1.414	˂0.001
No smoking or alcohol consumption	1.73 ± 1.312	1.24 ± 0.780	2.09 ± 1.618	1.12 ± 0.857	1.67 ± 1.348	˂0.001
Lack of psychological support from colleagues	1.61 ± 1.142	1.19 ± 0.657	1.76 ± 1.272	1.45 ± 1.081	1.56 ± 1.120	0.008

Note: the table shows the factors for which there exist statistically significant differences between the groups. The significance level of *p* is given according to these univariate criteria. *: The significance level is indicated according to tests of between-subjects effects of MANOVA

**Table 4 ijerph-19-00997-t004:** Statistically significant differences according to univariate tests in the subjective assessment of the discomfort of climate and geographic, production and social factors in FIFO personnel with a different fly-in period duration (on a five-point scale, where 5 is the most uncomfortable factor, 1 is the least uncomfortable one).

Factor Name	M ± SD for 2 Weeks	M ± SD for 1 Month	M ± SD for More Than 1 Month	M ± SD for Entire Sample	Significance Level *
Climate and geographic factors					
Low or high temperatures	2.47 ± 1.422	2.94 ± 1.526	2.74 ± 1.292	2.73 ± 1.447	0.038
High humidity	1.92 ± 1.299	2.10 ± 1.205	2.42 ± 1.417	2.11 ± 1.301	0.035
Change of time zones	1.16 ± 0.779	1.37 ± 0.866	1.60 ± 1.228	1.35 ± 0.949	0.006
Geomagnetic disturbances	1.15 ± 0.652	1.59 ± 1.080	1.68 ± 1.251	1.46 ± 1.022	˂0.001
Wind	2.21 ± 1.281	2.92 ± 1.451	3.44 ± 1.272	2.79 ± 1.429	˂0.001
Radiation situation	1.44 ± 1.045	2.07 ± 1.372	1.69 ± 1.300	1.76 ± 1.274	˂0.001
Barometric pressure drops	1.43 ± 1.009	1.68 ± 1.077	1.97 ± 1.308	1.66 ± 1.129	0.004
Change in oxygen content in atmospheric air	1.32 ± 0.884	1.81 ± 1.274	2.26 ± 1.390	1.74 ± 1.232	˂0.001
Production factors					
Noise	2.66 ± 1.393	2.67 ± 1.507	1.83 ± 0.636	2.55 ± 1.397	0.002
Vibration	2.44 ± 1.433	2.00 ± 1.313	1.53 ± 0.784	2.11 ± 1.337	˂0.001
Chemical factors	1.74 ± 1.245	2.55 ± 1.525	1.43 ± 0.813	2.07 ± 1.409	˂0.001
Illumination	1.65 ± 1.016	2.11 ± 1.311	1.55 ± 0.932	1.85 ± 1.175	0.002
Physical exercise	1.81 ± 1.084	2.17 ± 1.243	1.80 ± 0.687	1.97 ± 1.129	0.025
Elevated equipment surface temperature	1.68 ± 1.152	2.11 ± 1.361	1.58 ± 0.747	1.86 ± 1.227	0.007
Static electricity level	1.28 ± 0.861	1.57 ± 1.035	1.45 ± 0.677	1.44 ± 0.931	0.057
Neuropsychic overload	2.06 ± 1.270	2.15 ± 1.373	1.60 ± 0.841	2.04 ± 1.279	0.055
No professional support from colleagues	1.33 ± 0.925	1.63 ± 1.209	-	1.49 ± 1.095	0.030
Socially domestic factors					
Social isolation of the team	1.66 ± 1.128	1.86 ± 1.186	2.75 ± 1.459	1.98 ± 1.296	˂0.001
Unfavorable accommodation conditions	1.52 ± 1.042	1.83 ± 1.037	2.87 ± 1.224	1.94 ± 1.193	˂0.001
Lack of food	1.87 ± 1.255	2.14 ± 1.366	2.87 ± 1.454	2.20 ± 1.393	˂0.001
Difficulty with transport and communication	1.85 ± 1.203	2.34 ± 1.402	3.03 ± 1.505	2.31 ± 1.421	˂0.001
Psychophysiological and psycho-emotional discomfort	1.60 ± 1.062	1.89 ± 1.189	2.42 ± 1.439	1.90 ± 1.239	˂0.001
Frequency of work	1.57 ± 0.998	1.77 ± 1.033	2.48 ± 1.400	1.85 ± 1.159	˂0.001
Forced circle of contacts	1.63 ± 1.123	1.76 ± 1.127	2.38 ± 1.296	1.85 ± 1.195	˂0.001
Lack of developed infrastructure	1.69 ± 1.175	2.00 ± 1.272	2.75 ± 1.311	2.05 ± 1.304	˂0.001
Narrowing the zone of personal space	1.52 ± 1.123	1.86 ± 1.261	2.38 ± 1.384	1.85 ± 1.277	˂0.001
Information exhaustion	1.68 ± 1.206	1.71 ± 1.045	3.00 ± 1.125	1.98 ± 1.243	˂0.001
Lack of a developed cultural infrastructure	1.59 ± 1.127	2.04 ± 1.345	3.16 ± 1.441	2.12 ± 1.414	˂0.001
No smoking or alcohol consumption	1.19 ± 0.812	1.61 ± 1.234	2.59 ± 1.768	1.67 ± 1.348	˂0.001
Lack of psychological support from colleagues	1.30 ± 0.867	1.48 ± 0.977	2.13 ± 1.504	1.56 ± 1.120	˂0.001

Note: the table shows the factors for which there exist statistically significant differences between the groups. The significance level of *p* is given according to these univariate criteria. *: The significance level is indicated according to tests of between-subjects effects of MANOVA

**Table 5 ijerph-19-00997-t005:** Statistically significant differences according to univariate tests in the subjective assessment of the discomfort of climatic, geographical, production and social factors in FIFO personnel in various industries (on a five-point scale, where 5 is the most uncomfortable factor).

Factor Name	M ± SD for Diamond Mining	M ± SD for Oil Production	M ± SD for Construction	M ± SD for Entire Sample	Significance Level *
Climate and geographic factors					
Low or high temperatures	1.91 ± 1.097	2.99 ± 1.508	2.91 ± 1.422	2.73 ± 1.447	˂0.001
High humidity	1.63 ± 1.027	2.11 ± 1.302	2.32 ± 1.355	2.11 ± 1.301	0.001
Geomagnetic disturbances	1.19 ± 0.609	1.48 ± 1.068	1.55 ± 1.115	1.46 ± 1.022	0.054
Wind	1.91 ± 1.069	3.04 ± 1.456	3.01 ± 1.407	2.79 ± 1.429	˂0.001
Radiation situation	1.40 ± 0.954	1.67 ± 1.173	1.96 ± 1.414	1.76 ± 1.274	0.007
Barometric pressure drops	1.37 ± 0.832	1.78 ± 1.252	1.70 ± 1.143	1.66 ± 1.129	0.058
Change in oxygen content in atmospheric air	1.27 ± 0.687	1.83 ± 1.334	1.89 ± 1.300	1.74 ± 1.232	0.002
Production factors					
Vibration	2.42 ± 1.479	2.27 ± 1.359	1.81 ± 1.176	2.11 ± 1.337	0.004
Chemical factors	1.61 ± 1.100	2.40 ± 1.477	2.07 ± 1.442	2.07 ± 1.409	0.002
Illumination	1.49 ± 0.894	2.07 ± 1.246	1.88 ± 1.213	1.85 ± 1.175	0.007
Physical exercise	1.54 ± 0.823	2.18 ± 1.279	2.04 ± 1.088	1.97 ± 1.129	0.001
Elevated equipment surface temperature	1.46 ± 0.990	1.92 ± 1.249	2.04 ± 1.286	1.86 ± 1.227	0.007
Air humidity	1.39 ± 0.953	1.98 ± 1.308	2.03 ± 1.181	1.86 ± 1.204	0.001
No professional support from colleagues	1.21 ± 0.686	1.45 ± 1.086	1.78 ± 1.304	1.49 ± 1.095	0.006
Socially domestic factors					
Social isolation of the team	1.43 ± 0.908	1.96 ± 1.264	2.23 ± 1.392	1.98 ± 1.296	˂0.001
Unfavorable accommodation conditions	1.27 ± 0.750	1.91 ± 1.150	2.26 ± 1.260	1.94 ± 1.193	˂0.001
Lack of food	1.85 ± 1.294	2.08 ± 1.324	2.44 ± 1.444	2.20 ± 1.393	0.008
Difficulty with transport and communication	1.40 ± 0.818	2.62 ± 1.406	2.51 ± 1.482	2.31 ± 1.421	˂0.001
Psychophysiological and psycho-emotional discomfort	1.40 ± 0.780	1.94 ± 1.223	2.09 ± 1.357	1.90 ± 1.239	0.001
Frequency of work	1.51 ± 0.943	1.64 ± 1.006	2.14 ± 1.274	1.85 ± 1.159	˂0.001
Forced circle of contacts	1.51 ± 0.990	1.74 ± 1.128	2.07 ± 1.282	1.85 ± 1.195	0.003
Lack of developed infrastructure	1.51 ± 1.078	2.05 ± 1.276	2.30 ± 1.348	2.05 ± 1.304	˂0.001
Narrowing the zone of personal space	1.27 ± 0.770	1.81 ± 1.302	2.13 ± 1.357	1.85 ± 1.277	˂0.001
Information exhaustion	1.34 ± 0.770	1.86 ± 1.217	2.35 ± 1.304	1.98 ± 1.243	˂0.001
Lack of a developed cultural infrastructure	1.51 ± 0.975	1.82 ± 1.203	2.60 ± 1.551	2.12 ± 1.414	˂0.001
No smoking or alcohol consumption	1.24 ± 0.780	1.28 ± 0.991	2.13 ± 1.595	1.67 ± 1.348	˂0.001
Lack of psychological support from colleagues	1.19 ± 0.657	1.36 ± 0.867	1.85 ± 1.343	1.56 ± 1.120	˂0.001

Note: the table shows the factors for which there are statistically significant differences between the groups. The significance level of *p* is given according to these univariate criteria. *: The significance level is indicated according to tests of between-subjects effects of MANOVA

## Data Availability

Certificate of registration of the database 2021621448, 5 July 2021. Application No. 2021621308 dated 24 June 2021. Psychological safety and adaptability of oil and gas production workers in the shift organization of labor in the conditions of the Far North; Certificate of registration of the database 2021621433, 1 July 2021. Application No. 2021621309 dated 24 June 2021. Psychological safety and adaptability of the builders of the “Crimean Bridge” object in the rotational organization of labor.

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
