# Peer review of "The Adverse Environmental Impact Factors Analysis on Fly-In-Fly-Out Personnel at Industrial Enterprises"

_ijerph, 2022, doi:10.3390/ijerph19020997_

Round 1

Reviewer 1 Report

Overview

All my comments are intended to improve some aspects of your paper. It is an interesting and relevant investigation. It is exceptionally documented and its objectives are clear. The object of study is specific and well defined.

That said, some considerations in the different sections that I encourage you to review:

Introduction

  • Explain better what is the fly-in-fly-out employment at the beginning of the paper. A specific definition of the concept is required.
  • It would be necessary to reference the ideas of lines 46 and 61.
  • It would be necessary to specify which are the "various diseases" observed in line 94.
  • Line 21 of page 5: a point should be added in Rustamov et al.
  • From line 26 of page 6 there is a format error. The page margin is wider than before.
  • Some errors with the line break on page 7, line 28

Methods

  • This section requires a more in-depth explanation.
  • It would be necessary to explain in more detail the instrument used. In particular, the scales used. Are these scales validated? What are its alpha indices?
  • The hypotheses should be in the introductory section.
  • The sample used should be specified in the methods section.
  • How was the data collected?
  • It would be more understandable to generate headings in this section to facilitate understanding: sample, procedure, analysis, instruments ...
  • What is the statistical normality test used? Explain in more detail

Results

  • It gives the impression that no ex post tests have been calculated in the MANOVAs. This will make it possible to determine between which groups there are differences when there are more than two groups subjected to analysis.

Thanks for your excellent work. I hope these notes are useful.

Author Response

Dear reviewer,

Thank you very much for your time and a positive attitude towards research!

All comments have been corrected.

The table below provides comments on what changes have been made.

Best regards, the author

Reviewer 2 Report

Research is important and its results very interesting, however its presentation can be improved in the article that is submitting to the Journal. Here I suggest some changes and additions that may be useful: In the title of the paper it is necessary to indicate that the study refers to industrial enterprises.

  1. The paper does not indicate that all the industrial enterprise studied and their environmental impact. It does not emphasize the antiquity of adverse environmental impact.
  2. Objectives are not clear (see line 55-83)
  3. Literature review (see 2.1. Climate...: line 110-115, 2.2 industrial...: line 188-243, and 2.3. Social...: line 312-361).
  4. Materials and Methods (you can add more details 4.1. Design, 4.2. Sample and Procedure, 4.3. Measures, 4.4. Data analysis).
  5. Results are well-written.
  6. Discussion is too short. You should discuss with the main findings (see line 762- (1. Why are positive or negative results? What value means? and How practical implications?). * Move your limitation into Conclusion and then added practical implications/contributions.
  7. Conclusion: concluded with the main findings (see 830-851).

Author Response

(The authors gave the same response as above.)

Round 2

Reviewer 2 Report

The paper is a good and meaningful revision. The paper is thoughtful and essentially accessible to its reading and reader of the journal.